# Chemical Constituents of *Cassia abbreviata* and Their Anti-HIV-1 Activity

**DOI:** 10.3390/molecules26092455

**Published:** 2021-04-23

**Authors:** Xianwen Yang, Zhihui He, Yue Zheng, Ning Wang, Martin Mulinge, Jean-Claude Schmit, André Steinmetz, Carole Seguin-Devaux

**Affiliations:** 1Laboratory of Cellular and Molecular Oncology, Luxembourg Institute of Health, L-1445 Luxembourg, Luxembourg; yangxianwen@tio.org.cn (X.Y.); yue.zheng@path.utah.edu (Y.Z.); nwangmd@yahoo.com (N.W.); asteinme@pt.lu (A.S.); 2Key Laboratory of Marine Biogenetic Resources, Third Institute of Oceanography, Ministry of Natural Sources, 184 Daxue Road, Xiamen 361005, China; hezhihui@tio.org.cn; 3Department of Infection and Immunity, Luxembourg Institute of Health, L-4354 Esch-sur-Alzette, Luxembourg; mmulinge@uonbi.ac.ke (M.M.); jean-claude.schmit@ms.etat.lu (J.-C.S.); 4Department of Biochemistry, School of Medicine, University of Nairobi, Nairobi P.O. Box 30197-00100, Kenya; 5Service National of Infectious Diseases, Centre Hospitalier de Luxembourg, L-1210 Luxembourg, Luxembourg

**Keywords:** *Cassia**abbreviata*, Fabaceae, anti-HIV, heterodimer, flavonoid

## Abstract

Three new (**1**–**3**) and 25 known compounds were isolated from the crude extract of *Cassia abbreviata*. The chemical structures of new compounds were established by extensive spectroscopic analyses including 1D and 2D NMR and HRESIMS. Cassiabrevone (**1**) is the first heterodimer of guibourtinidol and planchol A. Compound **2** was a new chalcane, while **3** was a new naphthalene. Cassiabrevone (**1**), guibourtinidol-(4α→8)-epiafzelechin (**4**), taxifolin (**8**), oleanolic acid (**17**), piceatannol (**22**), and palmitic acid (**28**), exhibited potent anti-HIV-1 activity with IC_50_ values of 11.89 µM, 15.39 µM, 49.04 µM, 7.95 µM, 3.58 µM, and 15.97 µM, respectively.

## 1. Introduction

*Cassia abbreviata* is a small-to-medium-sized branched tree of the Fabaceae. It is widely spread in the tropics, especially in southeast Africa, with a long history in traditional medicine for the treatment of numerous conditions [1], such as headaches, diarrhea, constipation, some skin diseases, malaria, syphilis, pneumonia, stomach troubles, uterine pains, and gonorrhea [2,3]. Pharmacological studies indicated that *C. abbreviata* showed a broad spectrum of biological activities, including CNS depression [4], hypoglycemia [5], anti-AIDS [6], hepatoprotection [7], antioxidant [8], antibacterial [9], etc. Although some fatty acid compositions were analyzed from its seed oil by gas chromatography (GC) [10], while several dimeric and trimeric flavonoids were proposed on the basis of the UPLC–MS spectroscopic data [11], the chemical component investigation on *C. abbreviata* was seldom reported. Up to now, only a new flavan [12] and two novel trimeric proanthocyanidins [9] were isolated.

Recently, we screened several crude extracts from different plants of *Cassia* species and found that *C. abbreviata* showed potent anti-HIV-1 activity. Therefore, a systematic phytochemical investigation was carried out, which led to the isolation of three new (**1**–**3**) and 25 known (**4**–**28**) compounds (Figure 1). Herein, we report the isolation, structure, and anti-HIV-1 activity of these compounds.

## 2. Results and Discussion

Compound **1** showed a molecular ion peak at *m/z* 535.1590 [M + H]^+^ in its positive HRESIMS (Appendix A), corresponding to the molecular formula of C_29_H_26_O_10_. Its ^1^H and ^13^C NMR spectroscopic data in DMSO-*d*_6_ (Table 1) exhibited 29 carbon signals, consisting of one ABX [*δ*_H_ 6.12 (1H, d, *J* = 2.4 Hz, H-8); 6.15 (1H, dd, *J* = 8.4, 2.4 Hz, H-6); 6.40 (1H, d, *J* = 8.4 Hz, H-5); *δ*_C_ 102.0 (d, C-8), 108.3 (d, C-6), 118.6 (s, C-10), 129.5 (d, C-5), 155.3 (s, C-9), 155.8 (s, C-7)], one 1,4-disubsituted [*δ*_H_ 6.76 (2H, d, *J* = 8.5 Hz, H-3′,5′); 7.24 (2H, d, *J* = 8.6 Hz, H-2′,6′); *δ*_C_ 114.8 (d × 2, C-3′,5′), 129.3 (d × 2, C-2′,6′), 130.3 (s, C-1′), 157.1 (s, C-4′)] and one penta-substituted [*δ*_H_ 6.09 (1H, s, H-6″); *δ*_C_ 95.2 (d, C-6″), 97.4 (s, C-10″), 107.7 (s, C-8″), 151.7 (s, C-9″), 154.0 (s, C-5″), 155.4 (s, C-7″)] benzoic moieties, besides to one methyl [*δ*_H_ 0.97 (3H, s, Me-17″); *δ*_C_ 23.8 (q, C-17″)], two *sp*^3^ methylenes [*δ*_H_ 2.57 (dd, *J* = 17.7, 5.3 Hz, H-4″β), 2.67 (d, *J* = 17.7 Hz, H-4″α); 2.62 (dd, *J* = 19.0, 4.8 Hz, H-12″α), 2.98 (dd, *J* = 19.0, 11.6 Hz, H-12″β); *δ*_C_ 20.2 (t, C-4″), 31.1 (t, C-12″)], four oxygenated *sp*^3^ methines [*δ*_H_ 4.19 (d, *J* = 1.6 Hz, H-2″); 4.26 (br. t, *J* = 9.1 Hz, H-3); 4.39 (t, *J* = 2.5 Hz, H-3″); 4.48 (d, *J* = 9.5 Hz, H-2); *δ*_C_ 69.5 (d, C-3), 72.8 (d, C-3″), 78.7 (d, C-2″), 82.9 (d, C-2)], two *sp*^3^ methines [*δ*_H_ 2.36 (dd, *J* = 11.7, 4.6 Hz, H-11″); 4.40 (d, *J* = 9.1 Hz, H-4); *δ*_C_ 40.4 (d, C-4), 50.2 (d, C-11″)], one carbonyl (*δ*_C_ 174.5 s, C-13″), and one acetalic quaternary carbon (*δ*_C_ 115.6 s, C-15″).

In the heteronuclear multiple bond connectivity (HMBC) spectrum, a diagnostic ketal-γ-lactone moiety could easily be deduced according to correlations of H-2″ to C-3″/C-15″, H-3″ to C-10″, H-11″ to C-3″/C-13″, H_2_-12″ to C-13″/C-15″, and H_3_-17″ to C-11″/C-15″, which by further cross peaks of H_2_-4″ to C-5″/C-9″/C-10″ constructed the fragment of planchol A (**6**) [13]. Moreover, HMBC correlations of H-2 to C-1′/C-2′/C-9, H-3 to C-1′/C-2/C-4/C-10, and H-4 to C-5/C-9/C-10 and the large coupling constant of H-2/H-3 (^3^*J*_H2-H3_ = 9.5 Hz) could be used to establish another fragment of guibourtinidol [14]. These two fragments could be connected via C-4 and C-8″ by the key HMBC correlations of H-4 to C-7″/C-9″ (Figure 2). According to the large coupling constant between H-3 and H-4 (^3^*J*_H3-H4_ = 9.1 Hz), the stereochemistry of C-4 was assumed to be α-orientation [15]. On the basis of the above evidence and from the perspective of the biosynthetic pathway, the structure of **1** was then determined to be guibourtinidol (4α→8) planchol A, and named cassiabrevone.

Although dimers or trimers of flavonoids were commonly found in nature, cassiabrevone is the first example of a heterodimer formed by flavanol and planchol A. It might be biosynthesized from a natural chalconol by dehydration, oxidation, and final *endo-*attacking via neighboring group participation induced Friedel–Crafts reaction (Figure 3).

Compound **2** was assigned the molecular formula C_15_H_16_O_6_ from its positive HRESIMS at *m/z* 293.0947 [M + H]^+^. The ^1^H and ^13^C NMR spectroscopic data of **2** were very similar to those of epifiliferol [16,17], except that an ABX aromatic ring instead of an AX benzoic moiety was found in **2**. The assumption was confirmed by the HMBC correlation of H_2_-4 (*δ*_H_ 2.75, 1H, dd, *J* = 16.2, 3.0 Hz, H-4a; 3.14,1H, dd, *J* = 16.2, 4.0 Hz, H-4b) to C-5 (*δ*_C_ 156.7 s)/C-9 (*δ*_C_ 131.7 d)/C-10 (*δ*_C_ 111.9 s). The small coupling constant between H-2 and H-3 (^3^*J*_H2/H3_ = 4.0 Hz) further confirmed the *erythro*-configuration of **2**. Accordingly, the structure of **2** was assigned as 9-dehydroxyepifiliferol. Interestingly, it might be originated by the same biosynthetic precursor as **1**, via a nucleophilic displacement reaction, followed by the oxidation reaction.

The molecular formula of compound **3** was assigned to be C_24_H_30_O_13_ according to its sodium adduct ion peak at *m/z* 557.1792 [M + H]^+^, suggesting ten degrees of unsaturation. The ^1^H and ^13^C NMR spectra (Table 1) of **3** were almost the same as those of cassiaglycoside II (**17**) [18], except that the *β*-d-glucopyranosyl moiety at the C-6 position was shifted to the C-2′ position. This was evidenced by the downfield shift from 74.9 to 79.8 of the C-2′ position. Further confirmation could be observed by the HMBC correlations of H-1″ (*δ*_H_ 5.23, 1H, d, *J* = 7.9 Hz) to C-2′ (*δ*_C_ 79.8 d) and H-1′ (*δ*_H_ 1H, 4.75, d, *J* = 7.7 Hz) to C-8 (*δ*_C_ 156.5 s). By detailed analysis of its HSQC, COSY, and HMBC NMR spectroscopic data, compound **3** was then elucidated as 6-deglucopyranosyl-2′-glucopyransoyl cassiaglycoside II, and named cassiaglycoside V.

By comparison of the NMR and MS data with those published in the literature, 25 known compounds were determined to be guibourtinidol-(4α→8)-epiafzelechin (**4**) [15], guibourtinidol-(4α→8)-epicatechin (**5**) [15], planchol A (**6**) [13], (+)-afzelechin (**7**) [19], taxifolin (**8**) [20], dihydrokaempferol (**9**) [20,21], naringenin (**10**) [22], rhusopolyphenol E (**11**) [23], cascaroside D (**12**) [24], 1″-deoxyaloin B-1-*O*-*β*-d-glucopyranoside (**13**) [24], 10-hydroxycascaroside C (**14**) [24], cassialoin (**15**) [24], chrysophanol (**16**) [25], oleanolic acid (**17**) [26], erythrodiol (**18**) [26], lupeol (**19**) [27], *β*-sitosterone (**20**) [28], *β*-sitosterol (**21**) [29], piceatannol (**22**) [30], markhamioside F (**23**) [31], vanillic acid (**24**) [32], cassiaglycoside II (**25**) [18], (*7S*, *8S*)-syringoylglycerol (**26**) [33], *β*-d-glucopyranosyl (1→2)-β-d-glucopyranoside (**27**) [34], and palmitic aicd (**28**) [35].

The crude extract of *Cassia abbreviata* and all isolated compounds were assessed for their anti-HIV activity in MT4 cells infected by the reference strain HIV-1 IIIB (Figure 3). Cassiabrevone (**1**), guibourtinidol-(4α→8)-epiafzelechin (**4**), taxifolin (**8**), oleanolic acid (**17**), piceatannol (**22**), and palmitic acid (**28**) inhibited HIV-1 infection at noncytotoxic concentration and showed IC_50_ values ranging from 3.58 to 49.04 µM (Table 2). Enfuvirtide and Plerixafor are entry inhibitors for positive controls.

## 3. Materials and Methods

### 3.1. General Experimental Procedures

NMR spectra were recorded on Bruker 500 MHz spectrometer using TMS as an internal standard. The HRESIMS spectra were measured on a Waters Xevo G2 Q-TOF mass spectrometer. Optical rotations were measured with an Anton Paar MCP100 polarimeter. Chemical shifts were recorded in *δ* values using solvent signals DMSO*-d_6_* (*δ*_H_ 2.50/*δ*_C_ 39.5) and CD_3_OD (*δ*_H_ 3.0/*δ*_C_ 49.0) as references. Column chromatography (CC) was performed on silica gel, Sephadex LH-20, and ODS (octadecyl silane).

### 3.2. Plant Material

The mature shrubs of the plant (3 kg) were collected in Makueni County, Kenya, and the identity of *Cassia abbreviata* was confirmed by DNA barcoding.

### 3.3. Extraction and Isolation

Barks and roots of *Cassia abbreviata* were pulverized and extracted with 95% EtOH four times at room temperature. The extracts were combined and concentrated to provide a crude extract (CE, *ca* 420 g). Then, it was suspended in deionized water and partitioned successively with CHCl_3_, EtOAc, and *n*-BuOH, to provide an EtOAc-soluble extract (306.7 g) and an *n*-BuOH-soluble extract (103.5 g). The EtOAc extract was subjected to CC over silica gel eluting with a gradient CHCl_3_-MeOH (0→100%) to obtain eight fractions (Fr.1–Fr.8). Fr.2 was further separated by ODS and Sephadex LH-20 chromatography, and finally obtained **16** (58.1 mg) and **19** (16.8 mg) by preparative thin-layer chromatography (prep. TLC). Compounds **20** (32.7 mg) and **21** (68.5 mg) were purified from Fr.3 by repeated ODS CC, Sephadex LH-20 chromatography, and prep. TLC. Fr.5 was subjected to ODS CC, Sephadex LH-20 chromatography, and prep. TLC to give **9** (9.3 mg), **10** (11.5 mg), **11** (5.1 mg), **17** (0.8 mg), **18** (3.5 mg), **24** (11.5 mg), and **28** (6.9 mg). Purification of Fr.6 by ODS and Sephadex LH-20 chromatography, followed by prep. TLC led to the isolation of **1** (2.4 mg), **2** (3.4 mg), **4** (7.6 mg), **5** (6.5 mg), **6** (17.8 mg), **7** (40.9 mg), **8** (17.1 mg), **15** (140.3 mg), **22** (75.9 mg), and **26** (11.4 mg). The n-BuOH-soluble part was separated via ODS CC and Sephadex LH-20 chromatography, respectively. Final purification by prep. TLC obtained **3** (14.8 mg), **12** (20.0 mg), **13** (24.0 mg), **14** (22.0 mg), **23** (11.0 mg), **25** (21.0 mg), and **27** (28.7 mg).

Cassiabrevone (**1**): pale yellow amorphous powder; ${\left[\alpha \right]}_{D}^{20}$ −12.7 (*c* 0.1, MeOH); ^1^H and ^13^C NMR data, see Table 1; HRESIMS *m/z* 535.1590 [M + H]^+^ (calcd for C_29_H_27_O_10_, 535.1599).

9-Dehydroxyepifiliferol (**2**): white amorphous powder; ${\left[\alpha \right]}_{D}^{20}$ −12.0 (*c* 0.1, MeOH); ^1^H and ^13^C NMR data, see Table 1; HRESIMS *m/z* 293.1009 [M + H]^+^ (calcd for C_15_H_17_O_6_, 293.1020).

Cassiaglycoside V (**3**): pale yellow amorphous powder; ${\left[\alpha \right]}_{D}^{20}$ −90.0 (*c* 0.1, MeOH); ^1^H and ^13^C NMR data, see Table 1; HRESIMS *m/z* 557.1851 [M + H]^+^ (calcd for C_25_H_33_O_14_, 557.1865).

### 3.4. Anti-HIV-1 Infection Bioassay

MT4 cells were obtained through the NIH AIDS Reagent Program and cultured in RPMI 1640 (Lonza, Wijchen, the Netherlands) supplemented with 10% heat-inactivated fetal bovine serum (Lonza, the Netherlands) and 2mM l-glutamine (Invitrogen, Gosselies, Belgium). MT-4 cells were incubated with the crude extract of *Cassia abbreviatta* or the tested compounds alone to assess cytotoxicity, or HIV-1 IIIB alone or a mixture of the tested extract and compounds and HIV-1 IIIB viruses to assess protection against HIV-1 infection. After five days, protection from viral infection and the cytotoxicity were evaluated in parallel using (3-(4,5-dimethylthiazol-2-yl)-2,5-diphenyltetrazolium bromide (MTT, Sigma, Liège, Belgium) by measuring OD_540_ and OD_690_ using a POLARstar Omega Plate Reader (BMG Labtech, Ortenberg, Germany). Data were normalized to cells without treatment. Values of OD_540_−OD_690_ were calculated to determine IC_50_ values in Prism. The entry inhibitors, enfuvirtide, and AMD3100 (Sigma Aldrich, Liège, Belgium), were used as positive controls.

## 4. Conclusions

From *Cassia abbreviata*, three new compounds, cassiabrevone, 9-dehydroxyfiliferol, and cassiaglycoside V, were isolated along with 25 known ones. Noteworthily, cassiabrevone is the first heterodimer by flavanol guibourtinidol and tetracyclic phenolic planchol A. Moreover, six compounds showed inhibition against HIV-1 infection with IC_50_ values ranging from 3 to 50 µM.

## Figures and Tables

**Figure 1 molecules-26-02455-f001:**
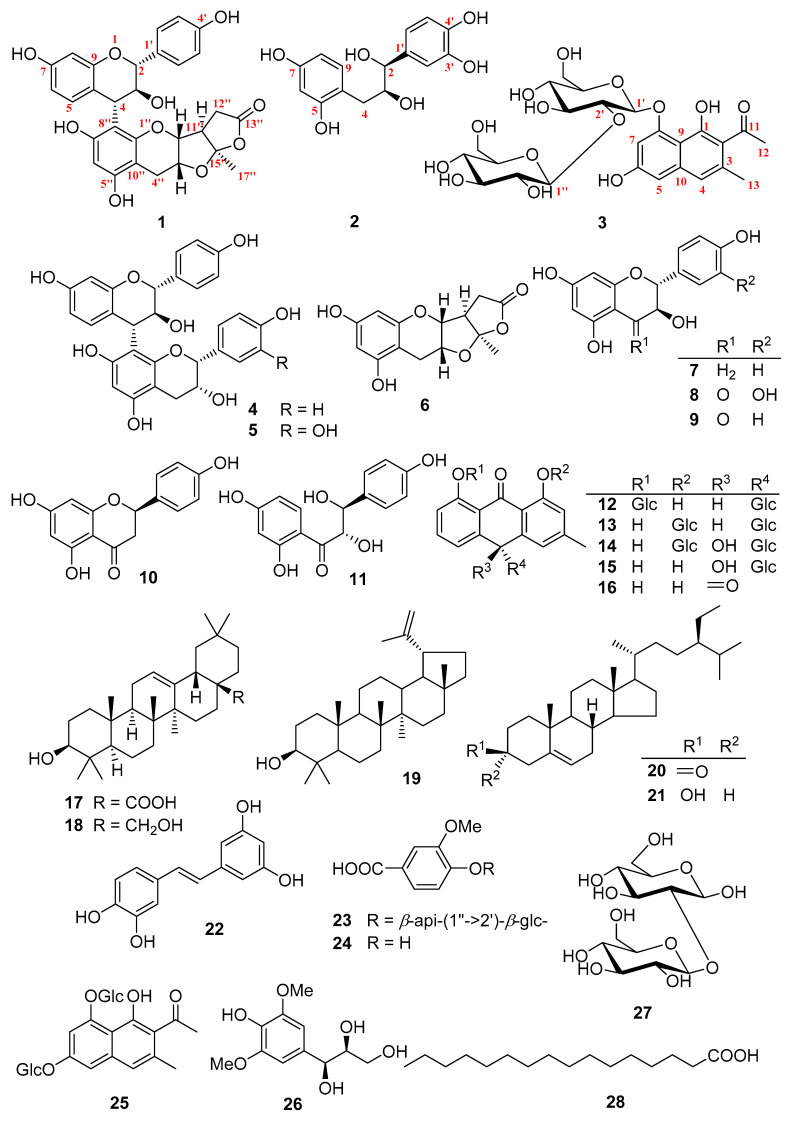
Compounds **1**–**28** from *Cassia abbreviata.*

**Figure 2 molecules-26-02455-f002:**
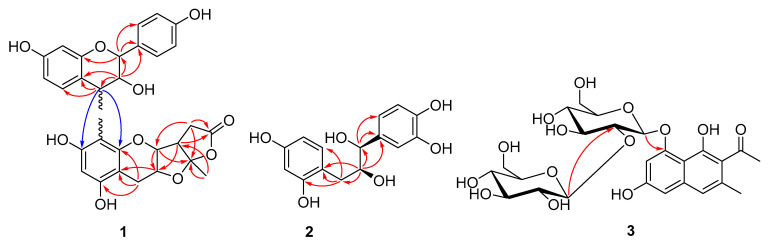
The key HMBC correlations of **1**–**3**.

**Figure 3 molecules-26-02455-f003:**
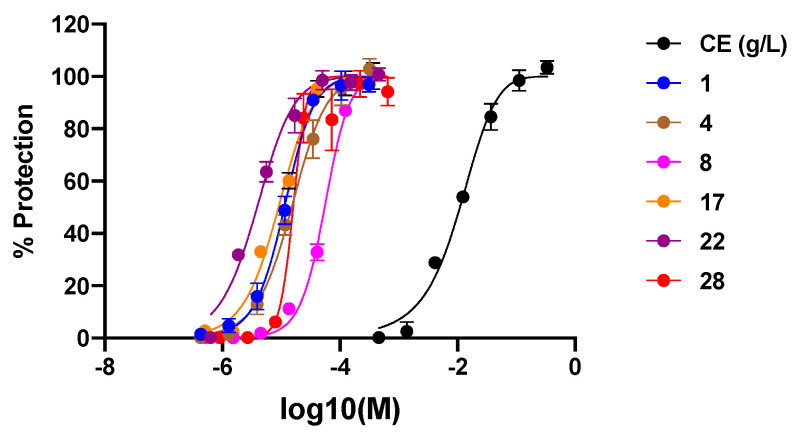
Protective effects of the crude extract (CE) along with compounds **1**, **4**, **8**, **17**, **22**, and **28** of *Cassia abbreviata* against HIV-1 infection (*n* = 3).

**Table 1 molecules-26-02455-t001:** ^1^H (500 Hz) and ^13^C (125 Hz) NMR spectroscopic data of **1**–**3** (*δ* in ppm, *J* in Hz within parentheses).

No.	1 ^a^		1 ^b^		2 ^b^		3 ^b^	
	*δ* _C_	*δ* _H_	*δ* _C_	*δ* _H_	*δ* _C_	*δ* _H_	*δ* _C_	*δ* _H_
1							153.7 C	
2	82.9 CH	4.48 (d, 9.5)	84.3 CH	4.56 (d, 9.6)	80.1 CH	4.90 overlap	123.3 C	
3	69.5 CH	4.26 (dd, 9.5, 9.1)	72.3 CH	4.40 (dd, 9.6, 9.1)	67.8 CH	4.19 (br. t, 4.0)	135.1 C	
4	40.4 CH	4.40 (d, 9.1)	41.8 CH	4.58 (d, 9.1)	34.1 CH_2_	3.14 (dd, 16.2, 4.2)	119.6 CH	6.86 s
						2.75 (dd, 16.2, 3.0)		
5	129.5 CH	6.40 (d, 8.4)	130.2 CH	6.60 (d, 8.6)	156.7 C		104.8 CH	6.64 (d, 2.0)
6	108.3 CH	6.15 (dd, 8.4, 2.4)	109.9 CH	6.24 (dd, 8.6, 2.4)	104.0 CH	(d, 2.4)	157.9 C	
7	155.8 C		157.0 C		157.8 C		102.7 d	6.74 (d, 2.0)
8	102.0 CH	6.12 (d, 2.4)	103.3 CH	6.24 (d, 2.4)	109.6 CH	(dd, 8.2, 2.4)	156.5 C	
9	155.3 C		156.9 C		131.7 CH	6.90 (d, 8.2)	109.3 C	
10	118.6 C		120.6 C		111.9 C		139.5 C	
1′	130.3 C		131.7 C		132.2 C		104.8 CH	4.75 (d, 7.7)
2′	129.3 CH	7.24 (d, 8.6)	130.3 CH	7.31 (d, 8.5)	115.4 CH	6.99 (d, 1.7)	79.8 CH	3.89 m
3′	114.8 CH	6.76 (d, 8.5)	116.0 CH	6.81 (d, 8.5)	145.9 C		78.0 CH	3.34 m
4′	157.1 C		158.5 C		146.0 C		78.2 CH	3.76 m
5′	114.8 CH	6.76 (d, 8.5)	116.0 CH	6.81 (d, 8.5)	116.0 CH	6.78 (d, 8.2)	71.0 CH	3.50 m
6′	129.3 CH	7.24 (d, 8.6)	130.3 CH	7.31 (d, 8.5)	119.4 CH	6.82 (dd, 8.2, 1.7)	66.9 CH_2_	3.23 m; 3.92 m
1″							100.4 CH	5.23 (d, 7.9)
2″	78.7 CH	4.19 (d, 1.6)	81.0 CH	4.22 (d, 2.4)			78.4 CH	3.54 m
3″	72.8 CH	4.39 (dd, 5.3, 1.6)	75.0 CH	4.45 (ddd, 5.1, 2.4, 1.5)			78.0 CH	3.34 m
4″	20.2 CH_2_	2.67 (d, 17.7)	21.3 CH_2_	2.89(d, 17.9)			75.4 CH	3.25 m
		2.57 (dd, 17.7, 5.3)		2.64 (dd, 17.9, 5.1)				
5″	154.0 C		155.6 C				71.0 CH	3.50 m
6″	95.2 CH	6.09 s	96.3 CH	6.09 s			62.3 CH_2_	3.74 m; 3.93 m
7″	155.4 C		157.1 C					
8″	107.7 C		109.4 C					
9″	151.7 C		153.0 C					
10″	97.4 C		99.6 C					
11″(11)	50.2 CH	2.36 (dd, 11.7, 4.7)	52.1 CH	2.39 (dd, 11.6, 4.5)			208.3 C	
12″(12)	31.1 CH_2_	2.98 (dd, 19.0, 11.7)	32.8 CH_2_	2.93 (dd, 19.2, 11.6)			32.7 CH_3_	2.58 s
		2.62 (dd, 19.0, 4.7)		2.60 (dd, 19.2, 4.5)				
13″(13)	174.8 C		177.2 C				20.2 CH_3_	2.23 s
15″	115.9 C		118.4 C					
17″	23.8 CH_3_	0.97 s	24.4 CH_3_	1.10 s				

^a^ Recorded in DMSO-*d_6_*. ^b^ Recorded in CD_3_OD.

**Table 2 molecules-26-02455-t002:** The IC_50_ values of compounds **1**, **4**, **8**, **17**, **22**, and **28** harboring an anti-HIV-1 activity.

Compounds	IC_50_ (µM)	
	HIV-1 Infection (µM)	Cytotoxicity
CE ^a^	9.98 ± 3.88 (µg/mL)	>1000 (μg/mL)
Cassiabrevone (**1**)	11.89 ± 2.14	>333
Guibourtinidol-(4α→8)-epiafzelechin (**4**)	15.39 ± 9.09	>333
Taxifolin (**8**)	49.04 ± 5.02	>333
Oleanolic acid (**17**)	7.95 ± 2.57	>333
Piceatannol (**22**)	3.58 ± 0.27	>333
Palmitic acid (**28**)	15.97 ± 3.04	>333
Enfuvirtide (T20) ^b^	0.0096 ± 0.001	>1
Plerixafor (AMD3100) ^b^	0.075 ± 0.009	>1

^a^ CE: crude extract of *Cassia abbreviate*. ^b^ Enfuvirtide and Plerixafor: positive controls. Each experiment was conducted three times and data were expressed as means ± SD.

## Data Availability

Not available.

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
