# Peer review of "Chemical Constituents of Cassia abbreviata and Their Anti-HIV-1 Activity"

_molecules, 2021, doi:10.3390/molecules26092455_

Round 1

Reviewer 1 Report

Dear authors,

please find enclosed the manuscript with my recommendations. 

Best,

Round 2

Reviewer 1 Report

Table 1 how do you explain different values for the coupling constant of H-5 and H-6? 

page 4

space character between "acetalic" and "quaternary"

Author Response

1. Thank you for your careful check. All coupling constant were calculated using the NMR frequency as 500 MHz, instead of 500.1330885 Hz, which caused the slight difference (0.1 Hz) between H-5 and H-6. In case of any misunderstanding, the authors changed it as the same in the revised manuscript.

2. Thank you. A space character was added between "acetalic" and "quaternary" in the revised manuscript.

Reviewer 2 Report

See attached file

Author Response

  1. The genus name was abbreviated without first appearance. (Please check all text again) Last time, the author had misunderstood for this meaning. L30, 36, 39 and other parts as C. abbreviata is OK.
    The authors are very sorry for the misunderstanding of the abbreviation of the genus name in last revision. They were corrected in this re-revised manuscript.
  2. Additionally the re-typing of this species name was mistyping "abbreviate" is wrong. Be careful!
    Thank you for the suggestion. It was corrected in the re-revised manuscript.
  3. L26 Fabaceae Family ---------> Fabaceae (this word is including family meaning)
    As suggested, “Fabaceae Family” was altered as “Fabaceae” in the re-revised manuscript.
  4. L64 acetalicquaternary ---------> acetalic quaternary
    Thank you. A space was added between “acetalic” and “quaternary” in the re-revised manuscript.
  5. L177 and CC50 ---------> (delete)
    CC50 was deleted in the re-revised manuscript.
  6. header of Table 2 like this (this is suggestion)
    The header of Table 2 was modified as suggested in the re-revised manuscript.